# Retrofitting of Imperfect Halved Dovetail Carpentry Joints for Increased Seismic Resistance

**Miloš Drdácký and Shota Urushadze \***

Institute of Theoretical and Applied Mechanics of the Czech Academy of Sciences, 190 00 Prague, Czech Republic; drdacky@itam.cas.cz

**\*** Correspondence: urushadze@itam.cas.cz; Tel.: +420-225-443-266

**Abstract:** This paper presents possibilities for anti-seismic improvement of traditional timber carpentry joints. It is known that the structural response of historical roof frameworks is highly dependent on the behavior of their joints, particularly, their capacity for rotation and energy dissipation. Any strengthening, or retrofitting, approach must take into account conservation requirements, usually expressed as conditions involving minimal intervention. Several retrofitting methods were tested on replicas of historical halved joints within various national and international research projects. The joints were produced with traditional hand tools, and made using aged material taken from a demolished building. The paper presents two approaches, each utilizing different retrofitting technologies that avoid completely dismantling the joint and consequently conserve frame integrity. The energy dissipation capacity is increased by inserting mild steel nails around a wooden pin, and connecting the two parts of the halved joint. In the second case, two thin plates made of a material with a high friction coefficient are inserted into the joint and fastened to the wooden elements. This is done by removing the wooden connecting pin and slightly opening a slot for the plates between the halved parts. In addition, the paper presents an application for disc brake plates, as well as thin plates made of oak.

**Keywords:** carpentry halved joint; energy dissipation; seismic retrofitting

## 1. Introduction

The behavior of carpentry joints in historical timber structures plays an important role in their overall structural response to applied loads, especially during seismic events. This has been the subject of several research projects, mostly in countries with high seismic activity, such as in the Mediterranean region. In these countries, roof framework joints employ a typical birdsmouth connection for joined timber elements, as shown in Figure 1. For example, Parisi and Piazza [1] tested the rotational capability of such carpentry joints retrofitted with various metal connectors. They have also modeled friction joints in traditional timber structures [2]. Parisi and Cordié [3] further studied the elastic rotational stiffness and post-elastic behavior of double-step timber joints, and the reversed birdsmouth configuration.

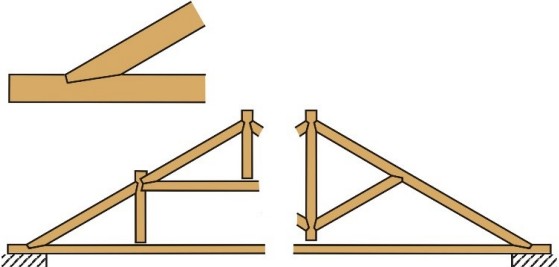

**Figure 1.** Typical birdsmouth joint and Mediterranean roof structure schemes.

Branco et al. [4] performed a series of tests on joints subjected to static and repeated loading to assess the impact of various strengthening methods. Palma et al. [5] published an extensive review of references, as well as the results of their own experimental research in rotational behavior of rafter and tie beam connections, including a study on the efficiency of some typical strengthening techniques. Similarly, Poletti et al. [6] studied traditional timber joints under cyclic loading, and possibilities of their repair or strengthening. All these works, for the most part, present techniques that are barely applicable to the structural restoration of historic timber structures.

Low slope roofs are typical in the Mediterranean region, in contrast to Central and Northern Europe, where steeper roofs are common and the roof framework structures use different details, especially halved dovetail joints with a higher rotational stiffness. Real joints in historical structures have many imperfections, for reasons ranging from low-quality carpentry work to material defects and degradation. These imperfections substantially influence the stiffness of roof trusses [7], decrease the safety of historical structures, and worsen the response of the roof frames in the event of seismic loading. A study of ways to improve the energy dissipation of carpentry joints was, therefore, carried out within the EC 7th Framework Programme NIKER project, which aimed to establish new integrated knowledge-based approaches for the protection of cultural heritage from earthquake-induced risks.

The performance of roof framework systems under seismic or generally dynamic loading has scarcely been studied even though roofs are sensitive to horizontal loads, especially those perpendicular to the plane of the trussed frame [8], despite the importance of such research having been underlined [9]. The stable roof framework as a three-dimensional box, and its potential role in a seismic event, has been analyzed by Giurani and Marini [10] but, again, only for low slope roofs. Steep roofs have not usually been considered for seismic loadings. However, they are typically constructed in a much stiffer way than low slope roofs, and do influence the building's performance. Their role in this respect has not been appropriately studied, and this is also not an aim of this paper.

With regard to historical structures, conservation requirements constrain retrofitting approaches to minimize interventions. Intervention should not significantly alter the appearance and behavior of a structure, and solutions that do not involve the complete disassembly of existing structures are preferred. This is important in the case of structural restoration works.

During an experimental campaign, a number of retrofitting processes were adopted and tested on replicas of historical halved joints that were made from old wood taken from a demolished building and produced using traditional carpenters' tools and techniques. Presented, here, are two approaches that each utilized a different retrofitting technique.

## 2. Experiments

### 2.1. Test Specimens

The experiments were carried out on replicas of traditional halved dovetail joints made from authentic timber (approximately 300 years old) taken from a demolished ancient building. The wooden elements had not been attacked by wood-destroying fungi or by wood-boring insects. Only drying fissures and cracks were present.

### 2.1.1. Material Characteristics

After the experiments on the joints were complete, the material characteristics of the spruce joint were determined by testing the standard coupons taken from the joint assemblies. The material properties evaluated were wood density and strength properties. Material information, mainly the coefficients of friction of the plates used for increasing energy dissipation, was taken from the literature [11]. The mechanical quantities of the brake plates were taken from the manufacturer [12]. The mechanical properties of the materials are summarized in Table 1.

**Table 1.** Material characteristics.

| Mechanical Property | Spruce | Oak | Brake Plate |
|---|---|---|---|
| Density—$\rho$ (kg/m³) | 340–450 | 700–750 | 1950 |
| Compression parallel to grain—$R$ (MPa) | 15.63 | – | – |
| Compression perpendicular to grain—$R$ (MPa) | 2.09 | – | – |
| Modulus of elasticity—$E$ (MPa) | – | 12,000 | – |
| Strength in compression—$f$ (MPa) | – | – | 180 |
| Coefficient of friction—$\mu$ | – | 0.41 * | 0.4 |

\* Friction coefficient of the oak parallel to the grain 0.48, perpendicular to the grain 0.34.

### 2.1.2. Geometrical Characteristics

The geometry of the joint assemblies varied, as did the extent of the wood's deterioration. An angle of 45° between the joined members was adopted. This is close to the angles most widely used in Central European roofing frames. The geometry of the specimens, the locations of the acting forces, and the potentiometer are depicted in Figure 2. The specimens were produced with intentional imperfections, as far as the tightness of the coupled elements was concerned, to model some degree of joint degradation. For this research, only two types of imperfection were considered—perfect joints with a tight connection between joined short struts in the overlap, and imperfect joints exhibiting a slight loss of contact in the abutment of the joined elements. The imperfections were produced during high-quality carpentry work and, therefore, their range was limited to local defects of 3 mm maximum. The geometric imperfections exhibit rather high slippage during the change in direction of the loaded arm rotation, as shown in Figure 8 and Figure 9, compared to a perfectly made joint, e.g., as shown in Figure 13. The increase in overall deformation increment of a roof truss due to joint slippage decreases with an increase of slippage value. This dependence is not linear, and the greatest difference in the overall deformation has been analyzed only around the change from perfect to imperfect joint geometry [7].

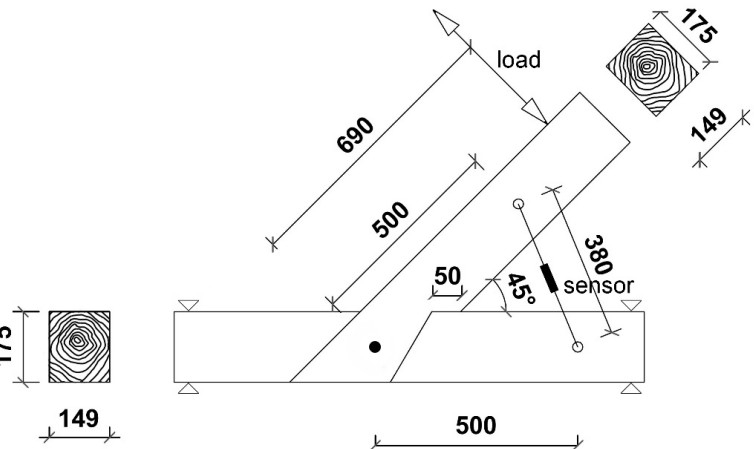

**Figure 2.** Geometrical scheme of the joint, the potentiometer location and location of the acting force, the direction of the joint's rotation, and the corresponding moment.

## 2.2. Joint Retrofitting

Several retrofitting techniques for increased energy dissipation were suggested and tested, as described above. However, for historic roof frameworks, only those which were acceptable from an aesthetic point of view were relevant for further development, especially in cases where the historic structures were exposed and accessible to visitors. Furthermore, the requirements of minimum intervention, re-treatability, and easy implementation were decisive for the selection of the tested techniques. Only the two most-suitable approaches for practical applications will be presented, as follows.

### 2.2.1. Metal Nails around a Wooden Pin

In the first case, energy dissipation capacity was increased by inserting mild steel rods (nails) around the wooden pin that connects the two parts of the halved joint, as shown in Figure 3a,b. No dismantling was necessary for this retrofitting intervention. The 6 mm diameter (D) nails were applied into pre-bored holes, which allowed for minimum spacing. When using six nails, the standard recommendations (10 × D along grains, 3 × D across grains) were observed. When using eight nails, due to the grain inclination, the distance along the grain was only 6 × D.

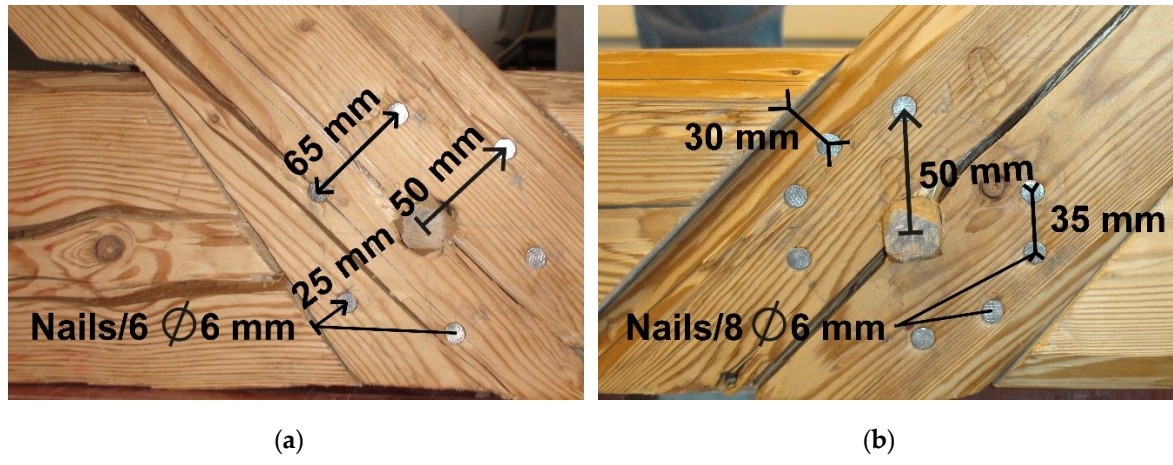

(**a**)  (**b**)

**Figure 3.** (**a**) Retrofitting using six mild steel nails; (**b**) retrofitting using eight steel nails.

### 2.2.2. Friction Joints

In the second retrofitting approach, the connecting wooden pin was removed from the joint, the halved parts were slightly separated, and two thin plates were then inserted into the opened slot and fastened to the wooden elements, as shown in Figures 4 and 5a,b. The plates were made of material with a high friction coefficient. Car disc brake plates and thin oak plates were used. The joint was then fixed and tightened with a steel bolt, which was pre-stressed to a certain degree. The screw bolt enables the joint's pre-stress level to change, which influences not only the stiffness of the joint, but also the frictional force between the plates. Several bolt pre-stressing moment values were chosen and tested (from 115 up to 240 Nm), which generated a stress level on the friction surfaces of around 0.43–0.90 MPa.

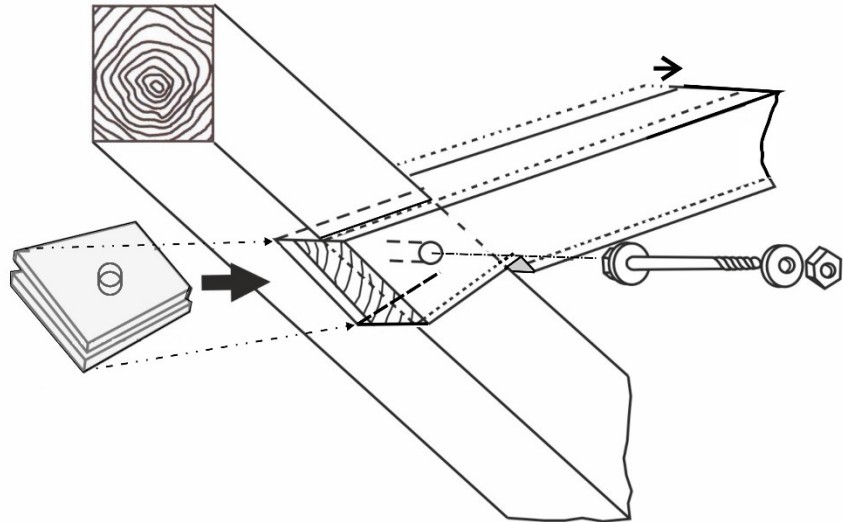

**Figure 4.** Schematic sketch showing insertion of friction plates into a halved dovetail joint, without complete dismantling, which is the main advantage of this approach.

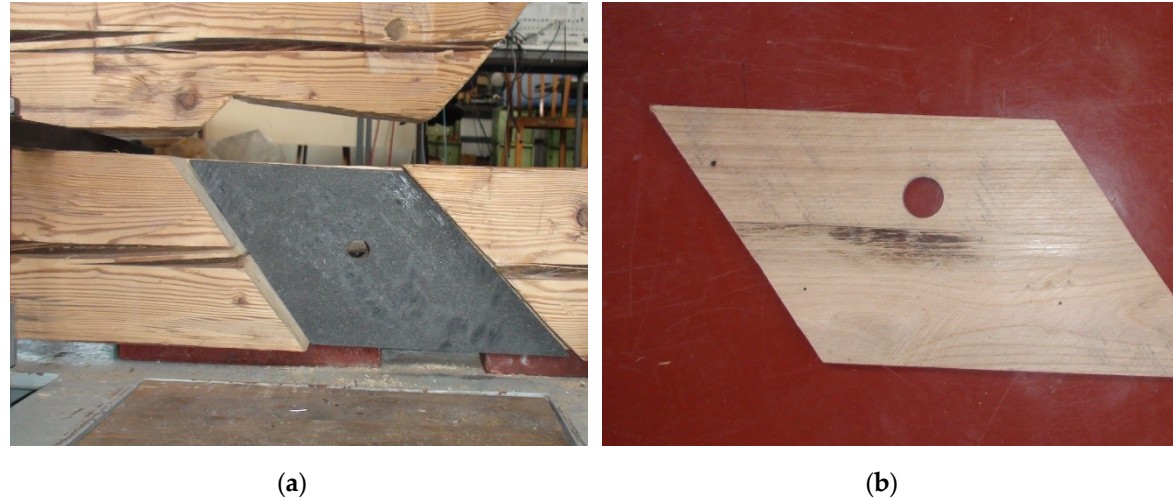

    (**a**)                                                      (**b**)

**Figure 5.** (**a**) Joint retrofitting with brake friction plate and replacement of the oak pin by a pre-stressed steel screw bolt; (**b**) an oak plate for joint retrofitting with friction plates.

### 2.3. Test Arrangement

On the basis of positive experience gained from previous tests performed at the Institute of Theoretical and Applied Mechanics of the Czech Academy of Sciences (ITAM) [7,13], a similar test set-up was constructed, as shown in Figure 6. This set-up enabled cyclic loading and pseudo-dynamic behavior of the halved dovetail joints to be simulated separately from the roof frame. The joint samples for testing were placed into a special testing rig that enabled pseudo-dynamic cyclic loading. It also ensured static stability of the samples and their responses in only the direction of loading. The cyclic load was applied using a servohydraulic MTS Systems Corporation actuator (cylinder) (MTS headquarters, 14000 Technology Drive, Eden Prairie, MN, USA, 55344) with a capacity of 25 kN, attached to a steel frame. The rotational responses of the joints were measured indirectly by means of a MEGATRON Elektronik GmbH & Co. KG SPR 18-S-100 (5 kΩ) potentiometer (MEGATRON Elektronik GmbH & Co. KG · Hermann-Oberth-Strasse 7 · 85640 Putzbrunn/Munich, Germany).

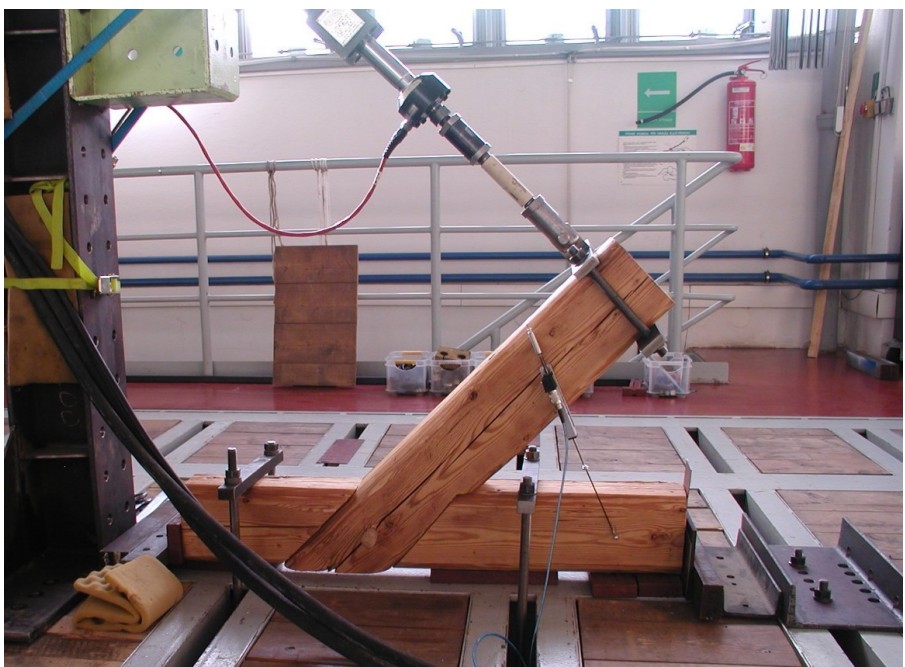

**Figure 6.** The test set-up.

The specimens were cyclically loaded, and the load deformation curves were registered. The load was applied to the joint using the actuator attached to the oblique beam. The intensity of the actuator's force was controlled by its prescribed displacement. The amplitude of the controlled displacement increased for each cycle, with a constant step equal to 4 mm. The frequency of each cycle was 0.1 Hz. During the tests, the forces needed to achieve the desired displacement of the cylinder and the change in the length of the potentiometer were recorded, as shown in Figure 7.

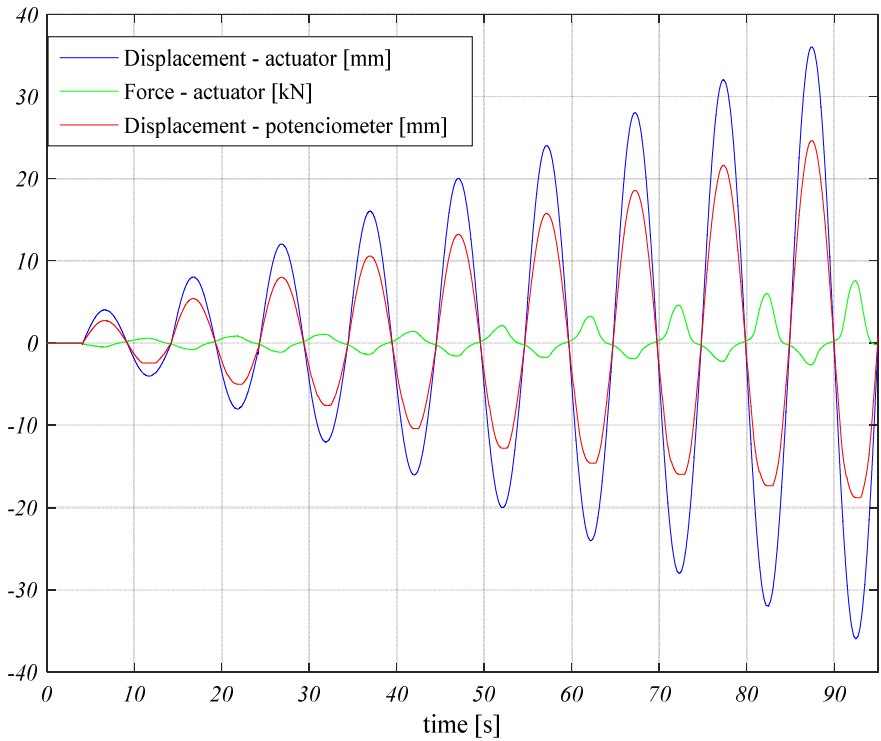

**Figure 7.** Time histories of the measured quantities during cyclic tests.

## 3. Results

Damping property is a fundamental factor influencing a structure's seismic resistance. Effective dissipation of energy by a structural element, e.g., a joint, can significantly decrease a structure's vibration level and lessen internal forces. The dissipative properties of the investigated halved dovetail joints can be described by the area of hysteresis loops, as shown in Figures 8a,b,9–11. Here, the hysteresis loops area representation of how the actuator's moment of force about the axis of pin M is dependent on the rotation of joint $\Delta\alpha$ for one loading cycle.

Both of the retrofitting methods tested (see Discussion) were effective from an energy consumption point of view. The results showing the changes in hysteresis loops after joint retrofitting are presented in Figures 8a,b,9–11. Figure 8b clearly shows an apparent increase in the energy needed to achieve the required rotation (displacement), in contrast to the unreinforced joint, represented in Figure 8a. In this case, the cyclic loading continued after the maximum testing load was reached, following the stability of the response loop. A negligible change in the hysteresis loop was observed.

Similar positive effects were attained in tests with the inserted friction plates. A typical example of the behavior of a joint with two brake plates inserted in the gap between overlapping parts of the joined elements is shown in Figure 10. As shown in Figure 11, an almost identical dissipation energy was achieved with the two oak plates (Figure 11), even though the pre-stressing forces applied to the joining bolts were not identical.

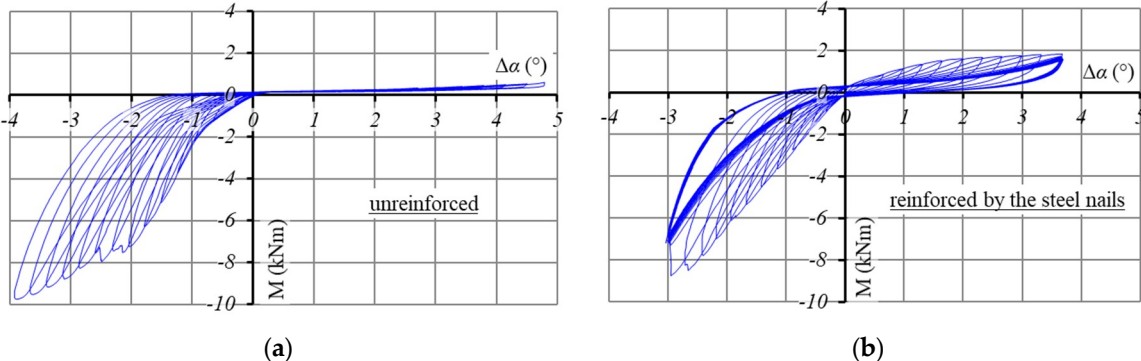

(**a**)  (**b**)

**Figure 8.** (**a**) Hysteresis loops of a joint before retrofitting with steel nails; (**b**) Hysteresis loops of a joint after retrofitting with steel nails.

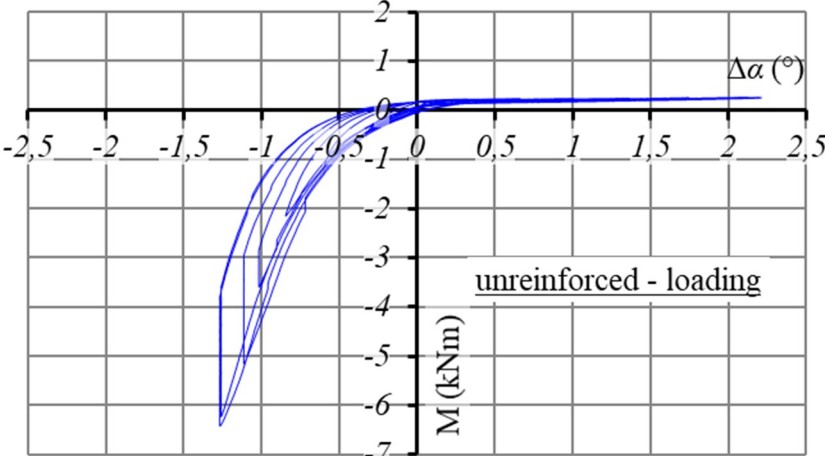

**Figure 9.** Hysteresis loops of a typical imperfect halved dovetail joint before retrofitting.

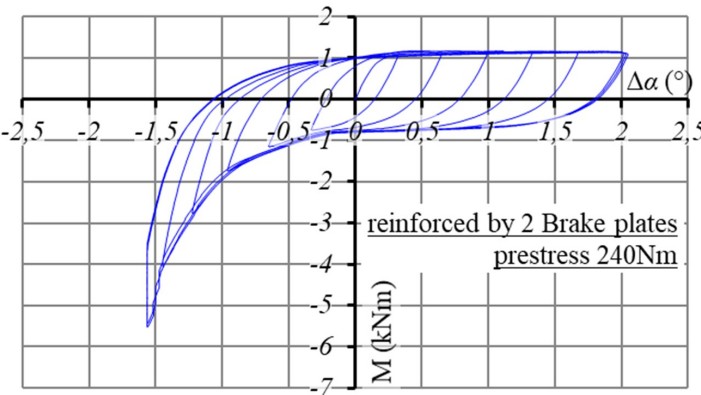

**Figure 10.** Hysteresis loops of a joint retrofitted with brake plates.

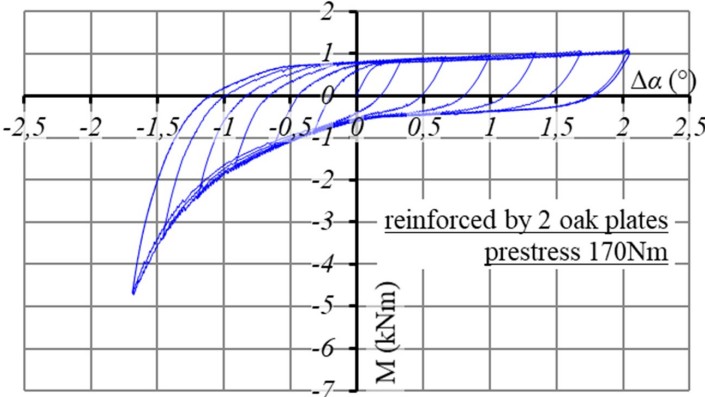

**Figure 11.** Hysteresis loops of a joint retrofitted with oak plates.

In the friction joints, energy dissipation depended on the forces pressing the two surfaces together. The resulting increase in energy dissipation, during cycling, of the tested joint retrofitting variations is shown in Figure 12. Comparing the results of the effect of two oak plates in relation to the brake plates—at an identical pre-stressing force—showed an obviously higher efficiency for the brake plates.

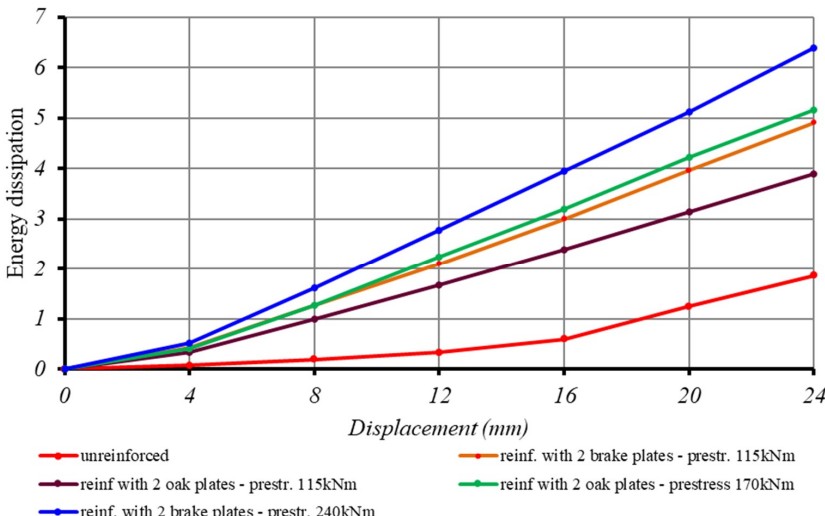

**Figure 12.** Energy dissipation curves for joints, both unreinforced and reinforced with various types of structural modifications.

## 4. Discussion

Rotational stiffness and slipping of joints were studied in detail by Drdácký et al. [7] for both a typical baroque and gothic roof framework on which halved dovetail joints were widely used. The results showed that tight, well-made joints with a reduced chance of slipping decreased the overall roof frame deformation under horizontal wind or earthquake loads, while also providing reasonably high rotational stiffness and capacity. Highly skilled carpenters were capable of producing flawless joints which were free of gaps between the individual elements.

Perfectly made joints may exhibit a sufficiently high rotational capacity, as shown in Figure 13, and as also shown by Wald et al. [13], and can be modeled in a rather simple way using a component modeling approach, e.g., after Vergne [14], which represents a good tool for the description joint behavior under repeated loading, as shown in Figure 14. In such a model, the joint was divided into components, which were represented by a force–deformation diagram. It is supposed that the dowel resists the shear force and clearly fixes the position of the connection's center of rotation. The results presented in Figure 14 show that more testing of the materials' characteristics are required, especially the behavior of wood under concentrated compression, to be able to describe the contact in the rafter dovetail indent. The tested composed joints are more complex and, therefore, an experimental investigation was adopted without any attempt to create computational models.

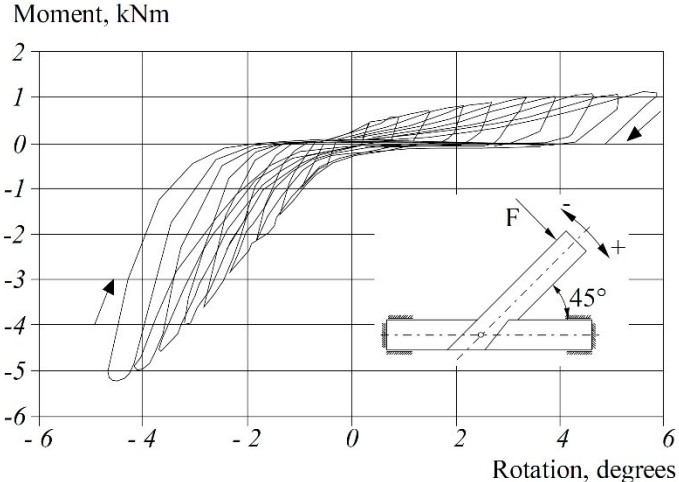

**Figure 13.** Example of an experimentally attained moment rotation diagram of a perfectly made halved dovetail joint [7].

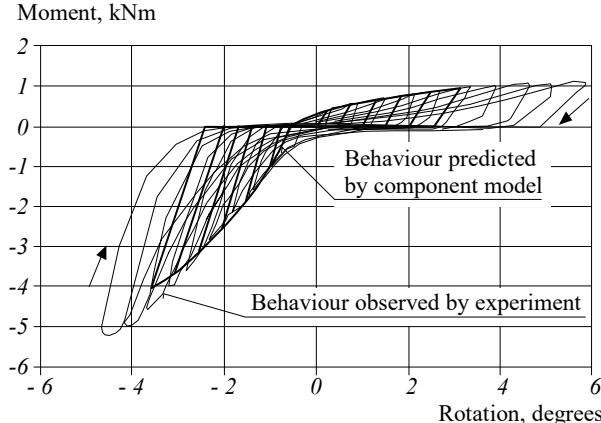

**Figure 14.** Comparison of the predicted component-based model for cyclic loading in the moment rotation diagram in Figure 13 [7].

Inserting nails around a wooden pin is a modification of the modern design of seismically resistant frame corners, Ceccotti A. [15], Kasal et al. [16]. The positive effect of this design has been published (for an example, Stehn and Börjes [17]). Our experiments have also demonstrated the effectiveness of this design, as shown in Figure 15a,b. The nails additionally strengthen and integrate the joint, and may replace a degraded wooden pin.

As the timber structure moves, the connected timber struts rotate mutually in the joint, and tend to deform the inserted steel rods plastically, which absorbs the energy, as shown in Figure 15b,c.

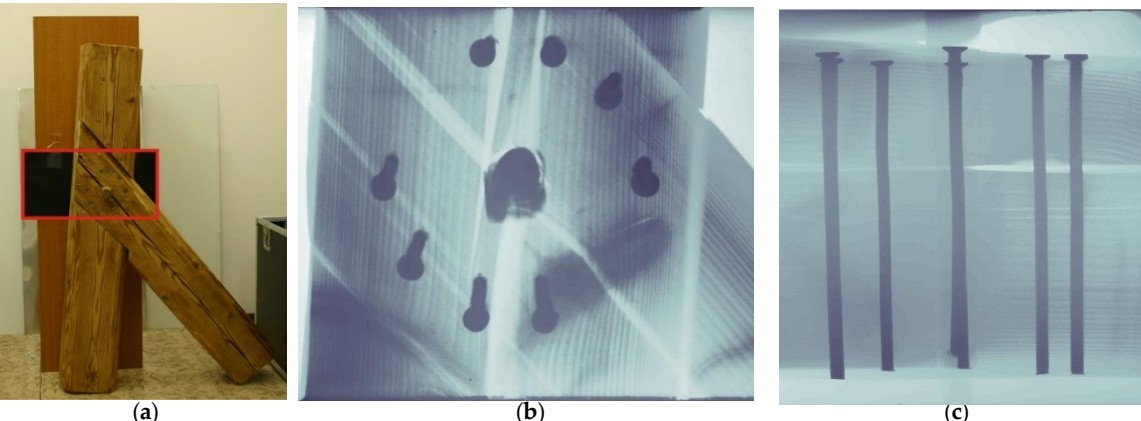

|     |     |     |
| --- | --- | --- |
| (a) | (b) | (c) |

**Figure 15.** X-ray presentation of the deformed mild steel nails after the cycling tests of a joint: the (**a**) X-ray recording, (**b**) perpendicular view of the joint plane, and (**c**) cross-sectional view of the joint plane.

A rather complex design, supported by appropriate numerical modeling, is required for an optimal interaction between nails and wood. The effect can be improved by inserting and fixing fiber-reinforced plastic (FRP) sheets on the contact surfaces of the halved joint counterparts. This requires partial dismantling of the joint, but prevents the origination and propagation of fissures or cracks.

Similarly, the approach based on improvement of friction between the contact surfaces of the halved joint requires moderate intervention, i.e., temporary removal of the wooden joining pin, and insertion and fixation of thin plates in the slot between the two joint components. The results are presented for an unreinforced joint, as shown in Figure 8; for brake plates, as shown in Figure 10; and for oak plates, as shown in Figure 11.

## 5. Conclusions

This experimental investigation focused on determining the effectiveness of several joint modifications with respect to their dissipative properties. The best results, and the highest level of effectiveness, were obtained for joints combined with plates having a high friction coefficient and a steel bolt. Tightening the nut on the bolt that is replacing the pin influences the pre-stress of the joint, i.e., the friction force between the plates. The frictional force increases in direct proportion to an increase in the degree of pre-stress. However, compressive deformation of the wood limits the maximum possible value of friction force. Fully fastening the brake plates to the wood using screws was the most effective method. Plates made of oak can be used as an alternative to brake plates, however, their damping efficiency is not as high. Reinforcing the joints with nails gave good results, as did the use of a combination of nails and inserted plates.

**Author Contributions:** M.D. and S.U. have done research and analyses, interpreted the results, as well as writes all sections of the paper. Original Draft Preparation, Writing-Review & Editing, M.D.

**Funding:** This study was supported by the EC FP7 Collaborative NIKER project (Grant agreement No. 244123) [18,19], the grant project DG18P02OVV040 "Monuments in motion", NAKI program II, provided by the Ministry of Culture of the Czech Republic and the ITAM institutional fund RVO 68378297.

**Acknowledgments:** The writers would like to thank master carpenter Stejskal for careful preparation of the test specimens, to Jaroslav Valach for X-ray photography and Marek Eisler for language and style correction.

**Conflicts of Interest:** The authors declare no conflict of interest.

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
