# Peer review of "Retrofitting of Imperfect Halved Dovetail Carpentry Joints for Increased Seismic Resistance"

_buildings, doi:10.3390/buildings9020048_

Reviewer 1 Report

This is a very interesting paper on traditional joints. Two different retrofitting techniques are presented, both showing a good seismic performance when compared to the unreinforced joint. The paper at this stage appears to be lacking some details and it would benefit from a more critical discussion of the results. Further comments are provided below: Section 2: More details are needed when describing specimens and materials. You mention that old timber is used (I assume it is spruce) and then you present a table with the properties of 3 materials. Only later it is understood that Oak and brake plates regard the materials used for the second retrofitting. It should be made clearer in the text.  How were the properties for Spruce and Oak obtained? You mention that intentional imperfections were produced. How did you control the level of imperfections? Was it always the same? Even a small variation in the gap can cause great differences in the experimental results. section 2.2: more details should be given on how the retrofitting was chosen, not only describing it. Doesn't the first solution with 8 nails provide too little distance among the nails? section 2.3: Which standard did you use to define the loading protocol? Was the potenciometer applied at the same level of the actuator. There is quite a difference in terms of displacement applied and measured. Please comment. Section 3: figures 8 to 11 should be thoroughly commented describing the changes observed in the hysteresis loops Same for figure 12, commenting both on level of pre-stressing and of type of plate. Section 4: did you apply the component model to your case? This part is rather confusing... Figure 14 needs to be explained better.

Author Response

This is a very interesting paper on traditional joints. Two different retrofitting techniques are presented, both showing a good seismic performance when compared to the unreinforced joint. The paper at this stage appears to be lacking some details and it would benefit from a more critical discussion of the results.

This criticism is reflected in answering detailed questions from reviewers.

Further comments are provided below:

Section 2: More details are needed when describing specimens and materials. You mention that old timber is used (I assume it is spruce) and then you present a table with the properties of 3 materials. Only later it is understood that Oak and brake plates regard the materials used for the second retrofitting. It should be made clearer in the text.

The material description has been expanded.

How were the properties for Spruce and Oak obtained?

It is described in the text and expanded.

You mention that intentional imperfections were produced. How did you control the level of imperfections? Was it always the same?

Included in the text. The imperfection was in a range of a high level carpentry work.

Even a small variation in the gap can cause great differences in the experimental results.

The relationship is not linear. The rise of an overall deformation increment due to the joint slippage decreases with an increase of the slippage value. The dependence is not linear and the greatest difference in the overall deformation of a roof truss has been analysed only around the change from the perfect to the imperfect joint geometry

section 2.2: more details should be given on how the retrofitting was chosen, not only describing it.

The text has been expanded.

Doesn't the first solution with 8 nails provide too little distance among the nails?

You are right – we accepted a slightly lower distance in the direction close to the direction along grains.

section 2.3: Which standard did you use to define the loading protocol?

No specific standard protocol was used.

RILEM Technical Committee 109 TSA, Timber structures in seismicregions. Dolan JD, editor. Mater Struct 1994;27(167):157–84.

Was the potenciometer applied at the same level of the actuator.  

The positions are clear from the Fig.2. The actuator was on in the same position as the potenciometer.

There is quite a difference in terms of displacement applied and measured. Please comment.

The difference is given by the different position of sensors – actuator LVDT and potenciometer.

Section 3: figures 8 to 11 should be thoroughly commented describing the changes observed in the hysteresis loops

The text has been expanded.

Same for figure 12, commenting both on level of pre-stressing and of type of plate.

The text has been expanded.

Section 4: did you apply the component model to your case? This part is rather confusing...

It is only presented as an example…

Figure 14 needs to be explained better.

The text has been expanded.

Thank you for very valuable comments.

Reviewer 2 Report

The investigate topic is very interesting for current earthquake engineering. In the last years, the topic has been widely investigated. Nevertheless, it remains an important issue in modern earthquake engineering and other studies are certainly useful for future applications. In particular, historical buildings and their seismic performances should still be widely investigated. It is opinion of the reviewer that each new study (if it's a good and rigorous work) could be considered for publication. In this way, the study could be considered for publication after several explanations, clarifications and improvements.

It is to be highlighted that the references should be expanded. In fact, several studies are not considered in the proposed study. Moreover, it is opinion of the reviewer that the literature review section is too long even more if it should be useful for the subsequent experimental part.

Likewise, the section 2 seems quite poor. Some more detailed description could be useful: for example, captions and labels to linked the mechanical properties (table 1) with figures.

Section 2.2. seems totally unuseful. In section 2.2, the authors say: “Several retrofitting techniques for increased energy dissipation have been suggested and tested. The two approaches most suitable for practical applications are presented.”. Really, no information are reported about several retrofitting techniques, tests, and results. Consequently, the considered approaches are no evaluable. In section 2.2.1, no information are reported about the used materials and details.

Section 4 does not seem a discussion.

The manuscript is rather short. Consequently, a comparison between the obtained experimental results and some analytical models could be considered. In fact, the reviewer (and maybe every reader) would like to know improvements and implication on the experimental and practice design.

Author Response

The investigate topic is very interesting for current earthquake engineering. In the last years, the topic has been widely investigated. Nevertheless, it remains an important issue in modern earthquake engineering and other studies are certainly useful for future applications. In particular, historical buildings and their seismic performances should still be widely investigated. It is opinion of the reviewer that each new study (if it's a good and rigorous work) could be considered for publication. In this way, the study could be considered for publication after several explanations, clarifications and improvements.

It is to be highlighted that the references should be expanded. In fact, several studies are not considered in the proposed study.

In fact the theme of the paper has not been so much studied. However, some more general references were added.

Moreover, it is opinion of the reviewer that the literature review section is too long even more if it should be useful for the subsequent experimental part.

It seems to be in contradiction with the comment above. We tried to do our best.

Likewise, the section 2 seems quite poor. Some more detailed description could be useful: for example, captions and labels to linked the mechanical properties (table 1) with figures.

The text on material tests has been expanded.

Section 2.2. seems totally unuseful . In section 2.2, the authors say: “Several retrofitting techniques for increased energy dissipation have been suggested and tested. The two approaches most suitable for practical applications are presented.”. Really, no information are reported about several retrofitting techniques, tests, and results. Consequently, the considered approaches are no evaluable.

The text on selection of techniques has been expanded.

In section 2.2.1, no information are reported about the used materials and details.

The text has been expanded.

Section 4 does not seem a discussion.

Various opinions are presented.

The manuscript is rather short.

The text has been expanded.

Consequently, a comparison between the obtained experimental results and some analytical models could be considered.

The work presents experimental results with a potential to be practically used without application of analytical modelling.

In fact, the reviewer (and maybe every reader) would like to know improvements and implication on the experimental and practice design.

Improvements and implications for practical design are presented.

Reviewer 3 Report

The manuscript discusses on two methodologies for seismic improvement of timber carpentry joints. The two investigated retrofitting aim to improve timber joint rotation capacity and energy dissipation.

It seems that the to two methodologies were investigated within a European research project (NIKER), but it does not result clear from the reading of the manuscript.

To some extent the subject matter of the paper is worthy of consideration, and the content of the manuscript is interesting. Nevertheless before to be considered for publication in a scientific paper several drawbacks must be addressed by the authors:

a) Introduction must include proper references that highlight/demonstrate the importance of i) capacity dissipation of timber joints and ii) stiffness of wooden floor/roof in the seismic capacity of buildings.

Few examples are listed here:

- Ferreira, T.M., Maio, R., Vicente, R. 2017. Analysis of the impact of large scale seismic retrofitting strategies through the application of a vulnerability-based approach on traditional masonry buildings. Earthquake Engineering and Engineering Vibration 16(2), pp. 329-348;

- Moreira, S., Ramos, L.F., Oliveira, D.V., Lourenço, P.B. 2016. Design Parameters for Seismically Retrofitted Masonry-To-Timber Connections: Injection Anchors. International Journal of Architectural Heritage 10(2-3), pp. 217-234;

- Betti M., Galano L., Vignoli A. 2014. Comparative analysis on the seismic behaviour of unreinforced masonry buildings with flexible diaphragms. Engineering Structures, 61 , pp. 195-208.

b) It must be clarified if the experimental tests were performed within the NIKER research project or not. The manuscript is confusing…

c) Description of the experimental tests, including the obtained results is poor, and must be enriched. It would be interesting to discuss the results reported in the manuscript with similar results obtained by other authors employing different techniques.

e) Not all the figures are quoted and discussed in the main text!

f) References section is not updated with the recent scientific advances. Authors should perform a critical literature search to look for recent papers on the subject.

Author Response

The manuscript discusses on two methodologies for seismic improvement of timber carpentry joints. The two investigated retrofitting aim to improve timber joint rotation capacity and energy dissipation.

It seems that the to two methodologies were investigated within a European research project (NIKER), but it does not result clear from the reading of the manuscript.

The role of the NIKER is clearly stated in the Introduction as well as in the acknowledgement.

To some extent the subject matter of the paper is worthy of consideration, and the content of the manuscript is interesting. Nevertheless, before to be considered for publication in a scientific paper several drawbacks must be addressed by the authors:

a) Introduction must include proper references that highlight/demonstrate the importance of i) capacity dissipation of timber joints and ii) stiffness of wooden floor/roof in the seismic capacity of buildings.

The Introduction has been expanded according to the suggestion.

Few examples are listed here:

The suggested examples are very far from the subject of the paper, however, some other references were added

- Ferreira, T.M., Maio, R., Vicente, R. 2017. Analysis of the impact of large scale seismic retrofitting strategies through the application of a vulnerability-based approach on traditional masonry buildings. Earthquake Engineering and Engineering Vibration 16(2), pp. 329-348;

- Moreira, S., Ramos, L.F., Oliveira, D.V., Lourenço, P.B. 2016. Design Parameters for Seismically Retrofitted Masonry-To-Timber Connections: Injection Anchors. International Journal of Architectural Heritage 10(2-3), pp. 217-234;

- Betti M., Galano L., Vignoli A. 2014. Comparative analysis on the seismic behaviour of unreinforced masonry buildings with flexible diaphragms. Engineering Structures, 61 , pp. 195-208.

b) It must be clarified if the experimental tests were performed within the NIKER research project or not. The manuscript is confusing…

The version of the paper for the OpenReview is reduced – the acknowledgement is excluded as well as the authorship. In the original manuscript the data are included.

c) Description of the experimental tests, including the obtained results is poor, and must be enriched. It would be interesting to discuss the results reported in the manuscript with similar results obtained by other authors employing different techniques.

The text has been expanded with relevant information. The existing results on the theme are included.

e) Not all the figures are quoted and discussed in the main text!

The text was corrected accordingly. Thank you.

f) References section is not updated with the recent scientific advances. Authors should perform a critical literature search to look for recent papers on the subject.

The references were expanded.

Thank you for your comments and suggestions.

Round  2

Reviewer 1 Report

The authors have adequately answered to the questions previously arisen and the quality of the paper has improved significantly, granting it publication.

Author Response

Thank you for very valuable comments.

Reviewer 2 Report

It is opinion of the reviewer that the manuscript has not been improved enough.

The literature review section was too long if compared with subsequent experimental part.

The authors say:

“The work presents experimental results with a potential to be practically used without application of analytical modelling.”

For this reason, the reviewer would like to see some comparison between the obtained experimental results and analytical models.

It is opinion of the reviewer that the improvements and implications for practical design are presented in strongly poor way.

Author Response

It is opinion of the reviewer that the manuscript has not been improved enough.

The authors are sorry not to satisfy all of this specific reviewer´s expectations, however, his requirements are so general and vague that a full satisfaction is rather difficult to achieve. We could continue the discussion in such a way ad infinitum. The other two reviewers consider the paper adequately improved and worth to be published, which proves that the opinions are very subjective and may differ in satisfaction.

The literature review section was too long if compared with subsequent experimental part.

The literature review section represents a fraction of the present version of the paper after the expansion which has been required by the other two reviewers. So again, the length is obviously a subjective parameter and the authors had to satisfy also other reviewers, moreover, they do not know any rule concerning the allowed length of the introductory part.

The authors say:

“The work presents experimental results with a potential to be practically used without application of analytical modelling.”

For this reason, the reviewer would like to see some comparison between the obtained experimental results and analytical models.

As it is stated in the paper, the study belongs to typical experimental works and it has not been aimed to present theoretical solutions. In fact, the main aim was to provide a technical solution ensuring minimum intervention into a heritage structure and thus to obey the strong requirements of conservation policy for structural restoration. This engineering approach is supported with an experimental evidence of a positive effect.

It is opinion of the reviewer that the improvements and implications for practical design are presented in strongly poor way.

The authors are very sorry to read such an assessment, however, they believe that this can be left for evaluation to the readers. The paper presents data useful for further studies or discussions.

Reviewer 3 Report

The authors provide a satisfactory response to the issues raised during the first round of review, and it is my opinion that the revised paper can be now considered for publication.

Author Response

Thank you for very valuable comments.

Round  3

Reviewer 2 Report

After review the authors have not satisfy reviewer´s evaluation.
On the contrary, they consider the manuscript adequately improved based on the other two reviewers evaluation (unknown a this reviewer)
For reviewer its well knonw that the study belongs to typical experimental works. Nevertheless, it is a good practice compare the obtained results with similar studies.
Moreover, the problem of the manuscript are not the theoretical solutions.
In fact, the study would like to provide a technical solution ensuring minimum intervention for structures. In this way, the authors do not seem to be aware that the current codes require specific formulation while proposed approach (althought supported by experimental evidence) is an empirical approach.
For this reason the results are not generalizable and immediately applicable

This manuscript is a resubmission of an earlier submission. The following is a list of the peer review reports and author responses from that submission.

Buildings EISSN 2075-5309 Published by MDPI AG, Basel, Switzerland RSS E-Mail Table of Contents Alert
Back to Top